# The Effect of Polymer Blends on the In Vitro Release/Degradation and Pharmacokinetics of Moxidectin-Loaded PLGA Microspheres

**DOI:** 10.3390/ijms241914729

**Published:** 2023-09-29

**Authors:** Hongjuan Zhang, Zhen Yang, Di Wu, Baocheng Hao, Yu Liu, Xuehong Wang, Wanxia Pu, Yunpeng Yi, Ruofeng Shang, Shengyi Wang

**Affiliations:** 1Key Laboratory of New Animal Drug Project, Gansu Province/Key Laboratory of Veterinary Pharmaceutical Development, Ministry of Agriculture and Rural Affairs/Lanzhou Institute of Husbandry and Pharmaceutical Sciences of CAAS, Lanzhou 730050, China; zhanghongjuan@caas.cn (H.Z.); yangzhen@caas.cn (Z.Y.); wudi@caas.cn (D.W.); haobaocheng@caas.cn (B.H.); liuyu@caas.cn (Y.L.); wangxuehong@caas.cn (X.W.); puwanxia@caas.cn (W.P.); yiyunpeng@saas.ac.cn (Y.Y.); 2Shandong Provincial Animal and Poultry Green Health Products Creation Engineering Laboratory, Institute of Poultry Science, Shandong Academy of Agricultural Science, Jinan 250023, China

**Keywords:** PLGA microspheres, moxidectin, polymer blends, in vitro release, pharmacokinetics

## Abstract

To investigate the effect of polymer blends on the in vitro release/degradation and pharmacokinetics of moxidectin-loaded PLGA microspheres (MOX-MS), four formulations (F1, F2, F3 and F4) were prepared using the O/W emulsion solvent evaporation method by blending high (75/25, 75 kDa) and low (50/50, 23 kDa) molecular weight PLGA with different ratios. The addition of low-molecular-weight PLGA did not change the release mechanism of microspheres, but sped up the drug release of microspheres and drastically shortened the lag phase. The in vitro degradation results show that the release of microspheres consisted of a combination of pore diffusion and erosion, and especially autocatalysis played an important role in this process. Furthermore, an accelerated release method was also developed to reduce the period for drug release testing within one month. The pharmacokinetic results demonstrated that MOX-MS could be released for at least 60 days with only a slight blood drug concentration fluctuation. In particular, F3 displayed the highest AUC and plasma concentration (AUC_0–t_ = 596.53 ng/mL·d, C_ave (day 30-day 60)_ = 8.84 ng/mL), making it the optimal formulation. Overall, these results indicate that using polymer blends could easily adjust hydrophobic drug release from microspheres and notably reduce the lag phase of microspheres.

## 1. Introduction

Poly (D, L lactic-co-glycolic acid) (PLGA), certified by the FDA, has been employed extensively for sustained-release drug delivery in the past few decades due to its attractive biocompatibility and biodegradable characteristics [1,2,3]. PLGA microspheres are one of the best studied long-acting drug delivery systems, as they can encapsulate a variety of drugs and the drug release rate can easily be tailored by regulating the polymer’s characteristics, such as molecular weight, terminal group and the ratio of two monomers [4,5,6,7]. PLGA microspheres display a variety of benefits over conventional drug delivery methods, including prolonged drug release duration, decreased dosing frequency, improved patient compliance, reduced toxicity and low medical costs [8]. There are several commercially available PLGA microspheres on the market, including Vivitrol^®^, Risperidal^®^, Lupron Depot^®^, Bydureon^®^ and Perseris™ [9,10,11,12,13]. Therefore, it is promising to develop PLGA microspheres for loading various therapeutic agents to achieve long-term drug delivery.

The desired long-acting injections should release the encapsulated drugs in nearly zero-order within the required time period. However, the in vitro release of drugs from PLGA microspheres typically involves three phases, including a burst release, followed by a long lag phase, and finally a near zero-order fast release [14,15]. The long lag phase during which only a very small amount of drug is released from microspheres is an important problem in the development of microspheres, as it could lead to insufficient blood drug concentration for treatment [16]. In order to eliminate the long lag phase, additives, for example porogens, are usually added to the formulation to modify release. However, the addition of additives may also result in local irritation and toxicity [17]. Hence, using polymer blending to prepare microspheres is a good strategy for adjusting drug release.

Many studies have reported using polymer blends to modify the release of microspheres [6,17,18,19,20]. Leuprolide acetate microspheres were prepared using PLGA polymer blends to reduce the burst release of microspheres [6]. By blending different PLGAs in various ratios, ganciclovir microspheres were prepared to effectively change the drug release rate of microspheres [17]. Accordingly, the lag phase caused by the slow degradation of high-molecular-weight PLGA could be minimized or even eliminated by combining high- and low-molecular-weight PLGA. However, there is a lack of a thorough investigation into the practical effects of polymer blends on the in vitro and in vivo performance of hydrophobic small-molecule PLGA microspheres.

In general, drugs are released from PLGA microspheres in two ways: diffusion and degradation/erosion [21,22]. The burst release of microspheres is usually controlled by diffusion, while the lag phase and subsequent fast release are controlled by the erosion of PLGA. There are four recognized processes of drug release from PLGA microspheres, including water absorption and swelling of microspheres, matrix destruction caused by hydrolysis of the polymer chain, internal pore formation and drug diffusion through pores. Therefore, understanding the in vitro release/degradation mechanism is crucial. In addition, a real-time release test at 37 °C is often time-consuming and cumbersome, making it necessary to develop an accelerated release method by changing the release condition.

In this study, we selected moxidectin (MOX), an anthelmintic macrocyclic lactone endectocide [23,24], as the model of a hydrophobic small molecule to evaluate the effect of polymer blends on the in vitro and in vivo performance of hydrophobic small molecule-loaded PLGA microspheres. For this aim, four moxidectin-loaded microspheres (MOX-MS) were prepared using various ratios of polymer blends and characterized for surface morphology, particle size, drug loading and the physical state of the drug within the microspheres. Thereafter, real-time in vitro release and accelerated release tests were performed, as well as an investigation into the in vitro degradation mechanism of microspheres. Furthermore, the pharmacokinetics of MOX-MS were evaluated. These results will provide a sufficient theoretical basis for controlling the release of microspheres by blending polymers so as to eliminate or minimize the lag phase.

## 2. Results and Discussion

### 2.1. Physicochemical Properties of MOX-MS

MOX-MS with various PLGA blends were prepared using the O/W emulsification solvent evaporation method. No significant differences were observed in the drug loading (36%) and encapsulation efficiency (91%) among four formulations (Table 1 and Figure 1), which may be explained by the high hydrophobicity of moxidectin. Nevertheless, with the increase of low-molecular-weight PLGA, the particle size reduced slightly (Appendix A), which might be attributed to the low viscosity of the organic phase during the preparation process caused by the addition of low-molecular-weight PLGA to the microspheres [25]. However, these results are contrary to those of previous studies about peptide drugs encapsulated in PLGA blends [18].

SEM results revealed that the surface of microspheres displayed irregular polygonal wrinkles with a diameter of about 1 μm (Figure 1 and Appendix A). This might be attributed to the high viscosity of the oil phase as well as the high hydrophobicity of the drug and PLGA, leading to the rapid solidification of the surface of the microspheres [26]. Moreover, four formulations also showed similar morphology and uniform particle size distribution, which was consistent with the results obtained by the laser diffraction particle size analyzer.

Using a powder X-ray diffraction technique, the existing state of moxidectin within the PLGA microspheres was analyzed. As illustrated in Figure 2, there was no obvious peak of moxidectin and PLGA, indicating that the drug existed in an amorphous state. In addition, there was no significant change in the FT-IR spectra of PLGA, moxidectin and moxidectin-loaded microspheres, indicating that there was no reaction between the drug and PLGA (Appendix A). In summary, these results are beneficial for the release control of microspheres [27].

The higher the surface hydrophilicity and free energy, the better the compatibility of microspheres. To evaluate the effect of different polymer blends on the surface wettability of microspheres, the water contact angles of different formulations were measured. The water contact angles were 74.13°, 69.40°, 63.20° and 54.70° for F1, F2, F3 and F4, respectively (Figure 3). This indicated that the addition of low-molecular-weight PLGA improved the surface compatibility of microspheres, which may be related to improving the wetting of microspheres and thus accelerating their in vitro release.

### 2.2. In Vitro Release of MOX-MS

#### 2.2.1. In Vitro Release of MOX-MS at 37 °C

Based on the solubility of moxidectin, a 10 mM PBS solution containing 0.5% SDS and 0.02% NaN_3_ was chosen as the in vitro release medium. All formulations showed a three-phase release, including a small burst release, a subsequent long lag phase, and finally an approximately zero-order fast release phase (Figure 4 and Appendix A). It was noteworthy that all formulations showed a long release period (the release durations of F1, F2, F3 and F4 were about 220 d, 180 d, 144 d and 108 d, respectively) and a small burst release. After the burst release on the first day (about 5%), a long lag phase of microspheres occurred, and the lag phase became shorter with the increase of the proportion of low-molecular-weight PLGA in microspheres (the lag phases of F1, F2, F3 and F4 lasted until days 144, 108, 70 and 49, respectively). As the release continued, an erosion-controlled release occurred, during which the drug was rapidly released at a near zero-order rate. In summary, the drug release cycle shortened with the elevated low-molecular-weight PLGA in the matrix. Moreover, after the completing release of the drug, the cumulative release decreased, which might be caused by drug degradation due to the long release period [28].

#### 2.2.2. In Vitro Release of MOX-MS at 50 °C/60 °C

Since the in vitro drug release from microspheres at 37 °C was time-consuming, it was crucial to develop an accelerated release method to facilitate quality control. It is reported that increasing the temperature is an effective method to accelerate the drug release from PLGA microspheres [14,26]. Therefore, the accelerated release studies were conducted by elevating the release temperature to 50 °C and 60 °C.

The release profiles of four formulations at 50 °C and 60 °C are showed in Figure 5A,B, respectively. At 50 °C, all formulations displayed a similar three-phase release to that observed at 37 °C [29]. The time for entire drug release was significantly shortened from 220 days to 28 days for F1, from 180 days to 24 days for F2, from 144 days to 22 days for F3 and from 108 days to 18 days for F4. In the first 24 h (Appendix A), only about 7% of the drug was released from microspheres, which was slightly higher than the percentage obtained at 37 °C. The lag phase and the third release phase of the four formulations were significantly shorter than those observed at 37 °C. At 60 °C, drug release from all formulations was further accelerated, with a higher burst release (15–18%), and the entire release cycle shortened to about 14 days(Appendix A). All formulations showed a nearly zero-order release profile, indicating that the drug release mechanism had changed dramatically. This may be due to the fact that 60 °C is higher than the glass transition temperature of all formulations, under which polymer fluidity increased and the degradation rate accelerated [30,31].

As listed in Table 2, the in vitro release of MOX-MS at 37 °C and 50 °C agreed with the sigmoidal equation. However, a zero-order kinetics model fitted best for the in vitro release of MOX-MS at 60 °C. Hence, the accelerated release at 50 °C might be the most suitable for quality control, as it was able to differentiate four formulations and significantly shorten the duration of the release test.

### 2.3. In Vitro Degradation Mechanism of MOX-MS

To analyze the in vitro degradation mechanism of MOX-MS prepared with various PLGA blends, at preset times, we monitored the matrix loss, the molecular weight of the PLGA, the pH of the surrounding bulk fluids and the morphology of microspheres.

#### 2.3.1. Mass Changes of MOX-MS

The in vitro release process was monitored and the remaining mass of the four formulations is shown in Figure 6A. In the first week of release, the weight of the microspheres of the four formulations changed slightly, with a mass loss of less than 5%. After incubation for 24 weeks, all formulations showed significant mass loss, and the remaining masses were 78.4%, 61.3%, 34.0% and 17.7% for F1, F2, F3 and F4, respectively (Appendix A-1). In addition, the kinetic fitting results showed that mass changes of MOX-MS in the degradation process agreed with a zero-order equation (Table 3). These results indicated that MOX-MS prepared using different polymer blends had different degradation rates, which correlated with the drug release rates from microspheres. It is speculated that polymer erosion played an important role in the release of moxidectin from PLGA microspheres.

#### 2.3.2. Polymer Mw of MOX-MS

The time-versus-Mw_t_/Mw_0_ plot (%, the Mw_t_ of PLGA t days after release/the Mw_0_ before release in vitro) is shown in Figure 6B. After 16 weeks of in vitro release, the Mw_t_/Mw_0_ values of F1, F2, F3 and F4 were 58.9%, 37.1%, 11.3%, and 1.9%, respectively (Appendix A-2). Compared with F1 prepared with 100% of high Mw PLGA (75/25, 75 kDa), the polymer blended formulations (F2–F4) showed a faster degradation rate. The polymer degradation rate substantially increased with the elevated proportion of low-molecular-weight PLGA (50/50, 23 kDa) in the microspheres. This could be explained by the fact that the PLGA in F1 exhibited high hydrophobicity and a slow degradation rate due to its high molecular weight and high LA/GA ratio (75/25, 75 kDa). Accordingly, the addition of low-molecular-weight PLGA to microspheres improved the water uptake and the subsequent polymer degradation. The mathematical model fitting results indicated that the polymer Mw changes of MOX-MS in the degradation process followed a zero-order equation (Table 3). This may have caused the mass changes of MOX-MS.

#### 2.3.3. pH Changes of Release Medium

Figure 6C illustrates the dynamic pH changes of the release medium upon exposing different PLGA microspheres to a 10 mM PBS solution (pH 7.4, containing 0.5% SDS and 0.02% NaN_3_) at 37 °C. Clearly, for F1, the pH value of the surrounding release medium changed slightly (from 7.59 to 6.94) during the 24 weeks of observation (Appendix A-3). For F2, the pH of the release medium changed slightly (from 7.67 to 7.10) in the first 16 weeks, but decreased rapidly (from 7.10 to 6.50) after 16 weeks. For F3, the pH of the release medium declined in the first 12 weeks, followed by a greater decrease in the next 12 weeks, and decreased to 4.63 by the 24th week. Compared to other formulations, F4 showed the fastest decrease rate of the pH value (from 7.61 to 4.72) from the 2nd week to the 16th week, while it decreased slowly after 16 weeks. These results indicate that the pH of the release medium decreased during the in vitro release process, and the decrease rate of pH increased with the elevated low-molecular-weight PLGA in microspheres.

In addition, mathematical model fitting results show that pH changes in the release medium of all formulations agree with near zero-order kinetics (Table 3). This indicates that the autocatalysis within microspheres and the potential acidification of the surrounding bulk fluids of the polymer caused by the leaching of PLGA degradation products from the particles seemed to play a significant role in the in vitro degradation process [32,33]. However, there was no linear relationship between the pH in the release medium and the in vitro release of MOX-MS, which may be due to the fact that the release of drugs from microspheres is the result of a combination of multiple mechanisms of action.

#### 2.3.4. MOX-MS Morphological Changes

The morphological changes of MOX-MS during in vitro degradation were also studied. According to Figure 7, the irregular polygonal wrinkles on the surface of microspheres slightly deepened after the first day of incubation, accompanied by drug diffusion on the surface of the microspheres. With the hydration of PLGA, the wrinkles on the surface of the microspheres gradually became shallower, which might be caused by the enhanced fluidity of the PLGA chains on the surface of microspheres in the release medium at 37 °C. As the microspheres continued to absorb water, the interior of the microspheres began to hydrolyze, and more and more pores formed on the surface and interior of the microspheres, accompanied by the swelling of the microspheres. The pores on F1, F2, F3 and F4 were first observed at 24, 16, 12 and 8 weeks, respectively. Meanwhile, the autocatalysis inside the microspheres accelerated the degradation of the microspheres, and the pores on the surfaces of the microspheres became larger, accompanied by the release of large amounts of drugs from the microspheres, which was consistent with the secondary erosion-accelerated release of microspheres [34].

The aforementioned in vitro degradation data indicate that the degradation of microspheres was from the inside to the outside, as reflected by the matrix loss, the molecular weight of the PLGA, the pH of the surrounding bulk fluids, and the morphology of microspheres during various release times [35]. The in vitro release process of MOX-MS is illustrated in Figure 8. In the first day of release, the drug on the surface of the microspheres diffused into the release medium [36,37], which caused the wrinkles on the surface of the microspheres to slightly deepen and formed an initial small burst release. With the further hydration of the microspheres, the fluidity of PLGA chains on the surface of the microspheres increased, the wrinkles on the surface of the microspheres gradually disappeared, and the surface of microspheres became smooth. The release of the microspheres began to enter a long lag period, during which only a small amount of drug was released from microspheres. Due to the autocatalysis inside the microspheres [38], the degradation was accelerated, accompanied by numerous pores formed inside the microspheres [39]. Random chain scission process took place, which significantly reduced the polymer Mw (Figure 6). As the degradation of the PLGA further accelerated, a large number of pores formed on the surface of microspheres. The acidic short chain degradation products of PLGA leached from the pores, and a second accelerated release phase started, during which a large amount of drugs was released from the microspheres.

To summarize, the release mechanism of MOX-MS consisted of a combination of pore diffusion and erosion mechanisms, in which autocatalysis played an important role in the degradation of microspheres. Adding of low-molecular-weight PLGA (10–50%) did not change the release mechanism of microspheres, but accelerated the release of the microspheres and shortened the lag phase.

### 2.4. Pharmacokinetic Evaluation

In order to assess the in vivo performance of MOX-MS prepared using various PLGA blends, the pharmacokinetics of the microspheres were investigated after subcutaneous injection into rats (1 mg/kg). Figure 9 and Table 4 show the moxidectin plasma concentration–time curves and the main pharmacokinetic parameters, respectively.

Moxidectin was rapidly absorbed after subcutaneous administration of moxidectin solution, reaching a peak concentration (C_max_) of up to 595.65 ng/mL at 0.04 d (Figure 9A and Appendix A). After administration for 1 d, the plasma concentration of moxidectin declined to 55.38 ng/mL. The half-life (T_1/2_) of the moxidectin solution was 8.66 d, suggesting that moxidectin was slowly eliminated in vivo, which was consistent with literature reports [40].

After subcutaneous administration of four different formulations made from various PLGA blends, the peak plasma concentrations (C_max_) of F1, F2, F3 and F4 were 44.49, 45.06, 28.30 and 27.25 ng/mL (Figure 9B), respectively, which were notably lower than those of the moxidectin solution group (595.65 ng/mL). The AUC_0-1d_ levels of F1, F2, F3 and F4 were 19.54, 19.91, 22.98 and 16.80 ng/mL, respectively, which were much lower when compared to the moxidectin solution group (135.85 ng/mL/d). Both Cmax and AUC_0-1d_ results indicated that MOX-MS significantly reduced the burst release. In addition, the AUC_0-t_ of F2 and F3 was much higher than that of the solution group, indicating that PLGA microspheres considerably improved the bioavailability of moxidectin.

Compared with the moxidectin solution group, MOX-MS exhibited a steadier plasma concentration level and a longer sustained release period. In the first month, the average plasma concentration of the moxidectin solution was higher than that of the microspheres group (Table 5). However, in the second month, the average plasma concentrations of F1, F2, F3 and F4 were 3.68, 5.13, 7.51 and 5.05 times higher than that of the moxidectin solution group, respectively. The microspheres still appeared to have a high plasma concentration, demonstrating that they had excellent in vivo release behavior. These results demonstrate that MOX-MS could provide an effective plasma concentration of at least 60 days in rats with a very small burst release. In particular, F3 showed the highest AUC (AUC_0-t_ = 596.53 ng/mL·d) and the steadiest plasma concentration, and therefore was considered as the optimal formulation. This also indicates that by regulating the proportions of PLGA with different molecular weights, microspheres with the desired in vivo release behavior might be produced.

## 3. Materials and Methods

### 3.1. Materials

Two types of poly (D, L lactic-co-glycolic acid) (PLGA) were obtained from Jinan Daigang Biomaterials: PLGA (LA/GA: 75/25, Mw: 75 kDa, carboxylic acid end group) and PLGA (LA/GA: 50/50, Mw: 23 kDa, carboxylic acid end group) (Jinan, China). Moxidectin was purchased from Jiangsu Lingyun Pharmaceutical Co., Ltd. (Changzhou, China). Polyvinyl alcohol (PVA, 87–89% hydrolyzed) was supplied by Kuraray Co., Ltd. (Osaka, Japan). Sodium dodecyl sulfate (SDS) was supplied by Shanghai Macklin Biochemical Co., Ltd. (Shanghai, China). Dichloromethane (DCM) and acetonitrile (ACN) were purchased from Fisher Scientific (Pittsburgh, PA, USA). All other chemicals and reagents were of analytical or chromatographic grade.

### 3.2. Preparation of Microspheres

Using different PLGA blends, four moxidectin-loaded microspheres (F1, F2, F3 and F4) were prepared using an oil-in-water (O/W) emulsion solvent evaporation technique [41]. Briefly, 900 mg of PLGA was dissolved in 6 mL of DCM, and 600 mg of moxidectin was dispersed in this solution. This organic phase was added slowly to 120 mL of 1% (*w*/*v*) aqueous PVA solution and homogenized at 4000 rpm for 2 min (IKA T25, Staufen, Germany) to form an O/W emulsion. Then, the resulting emulsion was transferred to 480 mL of purified water and stirred at 300 rpm for three hours to remove the solvent. Finally, microspheres were washed with purified water and then freeze-dried (SCIENTZ-10N, Ningbo SCIENTZ Biotechnology Co., Ltd., Ningbo, China). The obtained microspheres were stored at 4 °C in the dark until use.

### 3.3. Characterization of MOX-MS

#### 3.3.1. Particle Size and Size Distribution

A Mastersizer 2000 laser diffraction particle sizer was used to analyze the particle size and size distribution of microspheres (Malvern, Worcestershire, UK). Microspheres were uniformly dispersed in deionized water and dropped into the system for particle size analysis. D_50_ and Span (Span = (D_90_ − D_10_)/D_50_) were used to express the particle size and size distribution, respectively. All measurements were carried out in triplicate.

#### 3.3.2. Morphology

The particle morphology of MOX-MS was imaged using a scanning electron microscope (SEM) (TESCAN LYRA3 FIB-SEM, TESCAN, Brno, Czech Republic). Prior to SEM imaging, freeze-dried microspheres were sprayed with gold for 90 s under vacuum conditions.

#### 3.3.3. Drug Loading and Encapsulation Efficiency

According to USP 40, a high-performance liquid chromatography (HPLC) method was used to determine the concentration of moxidectin. Briefly, a Waters E2695 HPLC system (Waters, Milford, MA, USA) was equipped with a 2498 UV detector set at 242 nm. Analyses were performed using a C18 reversed-phase column (Nova-Pak^®^ C18, 3.9 × 150 mm, 4 μm) and an acetonitrile: buffer (60:40, *v*/*v*) mobile phase (the buffer was prepared by dissolving 7.7 g of ammonium acetate in 400 mL of water and adjusting with glacial acetic acid to a pH of 4.8). The column temperature was set at 50 °C and the flow rate was 2.5 mL/min.

Microspheres (20 mg) were dissolved in 10 mL acetonitrile, followed by filtering through a 0.45 μm PTFE membrane and HPLC analysis. The following equations were used to calculate the drug loading (*X*) and encapsulation efficiency (*E*) of MOX-MS:(1)$$X(\%)=\frac{W_{D}}{W_{M}}\times 100$$
where *W_D_* is the weight of moxidectin measured by HPLC after extraction of net-weight *W_M_* from MOX-MS.
(2)$$E(\%)=\frac{L_{A}}{L_{T}}\times 100$$
where *L_A_* is the actual moxidection loading, and *L_T_* is the theoretical moxidection loading.

#### 3.3.4. Powder X-ray Diffraction Analysis

Diffractograms of moxidectin, blank microspheres, the moxidectin and blank microspheres mixture, and MOX-MS (F1–F4) were obtained using an X-ray diffractometer (Ultima IV, Rigaku, Japan), using Cu Kα radiation. At a determined voltage (40 kV) and tube current (40 mA), samples were measured at a scan rate of 0.02°/min from a 2θ range from 2° to 60°.

#### 3.3.5. Surface Contact Angle

Four MOX-MS formulations were fixed on a DSA100 contact angle analyzer (KRUSS, Hamburg, Germany) to measure the water contact angle. Deionized water was used as the test fluid and the temperature and humidity were 25 °C and 65%, respectively. The measurement was repeated six times for each sample. The mean value θ was used as the final result.

### 3.4. In Vitro Release

The in vitro performances of MOX-MS were assessed using a sample-and-separation method [42]. Briefly, 20 mg of MOX-MS were suspended in 50 mL of release medium (pH 7.4, 10 mM PBS solution, containing 0.5% SDS and 0.02% NaN_3_). After that, all the samples were incubated at 37 °C in a water bath with a shaking speed of 100 rpm. At a determined time, 5 mL of supernatant was collected by centrifugation at 3000 rpm for 10 min, and filtered through a 0.45 μm membrane. The concentration of moxidectin in the medium was determined by HPLC. An equal volume of new release medium was supplemented at the same time. All the tests were performed in triplicate and the results were expressed as mean ± standard deviation. Accumulative drug release at each time point (*Q*_t_) was calculated according to the following equation:(3)$$Q_{t}(\%)=\frac{V_{0}\times C_{t}+V\times \sum_{n=1}^{t-1}C}{W\times X}\times 100$$
where *C*_t_ is the measured drug concentration at time t, *V*_0_ is the total volume of the release medium, *V* is the volume of each sampling, *W* is the total weight of microspheres initially used for the drug release test, and *X* is the drug loading of the microspheres.

To facilitate drug development and quality control, the accelerated in vitro release tests were carried out following the same procedure except that they were performed at elevated temperatures (50 °C and 60 °C) [43].

In vitro release results of different MOX-MS formulations were studied by fitting different kinetic models, such as zero-order or sigmoidal equations, using Origin 2022 software [17,44,45].

### 3.5. In Vitro Degradation Mechanism Studies

#### 3.5.1. Mass Changes of Microspheres

At predetermined time points (1 d, 1 w, 2 w, 4 w, 6 w, 8 w, 12 w, 16 w, 20 w and 24 w), microspheres were collected by centrifugation at 3000 rpm for 10 min, washed with purified water, and lyophilized. The freeze-dried microspheres were weighed. The remaining masses of microspheres after release for t days (*R*_t_) were recorded according to the following equation:(4)$$R_{t}(\%)=\frac{W_{t}}{W_{0}}\times 100$$
where *W*_0_ is the weight of the microspheres prior to release in vitro, and *W_t_* is the weight of the microspheres after release for t days.

#### 3.5.2. Morphological Characterization of Microspheres

At predetermined time points (1 d, 1 w, 2 w, 4 w, 6 w, 8 w, 12 w, 16 w, 20 w and 24 w), the collected freeze-dried microspheres were observed using a scanning electron microscope (TESCAN LYRA3 FIB-SEM, TESCAN, Brno, Czech Republic).

#### 3.5.3. Molecular Weight Analysis

Using gel permeation chromatography (GPC), the average molecular weight of the PLGA in the collected freeze-dried microspheres was measured (at 1 d, 1 w, 2 w, 4 w, 6 w, 8 w, 12 w, 16 w, 20 w and 24 w). A Waters 2515 HPLC pump, a Waters 2707 Plus Autosampler, a Waters 2414 refractive index detector and two Waters Styragel high-resolution columns (HT3 and HT4; effective molecular weight ranges: 500–30,00 and 5000–600,000, respectively) were installed on the instrument. The mobile phase was tetrahydrofuran (THF, HPLC grade) at a flow rate of 1.0 mL/min at 35 °C. Samples were dissolved in THF at a concentration of 1 mg/mL and filtered through a 0.45 μm filter before detection. Monodispersed polystyrene standards obtained from Waters Co. with a molecular weight range of 1000 to 2.0 × 10^5^ g/mol were used to generate the calibration curve. The weight-average molecular weight (Mw) and the number-weight molecular weight (Mn) were evaluated using Waters Millennium software [46,47].

#### 3.5.4. pH of Release Medium at Different Release Stages

To further reveal the in vitro degradation mechanism of microspheres, the release medium was collected to measure pH (FiveEasy Plus, Mettle Toledo, Zurich, Switzerland) at predetermined sampling points.

### 3.6. Pharmacokinetics Study

The pharmacokinetics of MOX-MS prepared with different PLGA blends were evaluated in rats. This study protocol was approved by the Ethics Committee of the Lanzhou Institute of Husbandry and Pharmaceutical Science of the Chinese Academy of Agricultural Sciences (Permit no. SYXK-2022-006). All animals were housed in clean stainless steel cages (3 rats per cage), with free access to food and water, and were maintained under 25 ± 2 °C conditions with a constant 12 h light–dark cycle. Before the experiment, animals were acclimatized for at least 7 days. In addition, During the experiment, all precautions were taken to minimize animal suffering.

Thirty male Sprague-Dawley rats (Lanzhou Veterinary Research Institute of Chinese Academy of Agriculture Science, Lanzhou, China), weighing 200–220 g, were randomly divided into five groups (*n* = 6). Groups 1, 2, 3 and 4 received F1, F2, F3 and F4, respectively, and Group 5 received a moxidectin solution (40 mg of moxidectin dissolved into 100 mL of 1% Tween 80 aqueous solution). MOX-MS were reconstituted with an aqueous medium (containing 0.87% sodium chloride, 0.1% Tween 20 and 0.75% sodium carboxymethyl cellulose) [12]. The five groups of rats were administrated drugs subcutaneously at a single dose of 1 mg/kg [48,49]. At predetermined sampling points, rats were placed in a glass anesthesia box containing infiltrated ether cotton balls until they were completely anesthetized. About 0.5 mL of blood was drawn from the orbital vein into heparinized centrifuge tubes. For the five groups, the sampling intervals were 1, 2, 4, 6, 8, 12 h and 1, 2, 3, 5, 7, 9, 12, 15, 20, 25, 30, 35, 40, 45, 50, 55 and 60 d. The supernatant plasma was collected by centrifuging the blood samples at 5000 rpm for 10 min at 4 °C, and then stored at −20 °C until further analysis. On the 61th day, all the animals were euthanized with ketamine hydrochloride 70 mg/kg (i.p.). The moxidectin concentration in plasma was analyzed by UPLC-MS/MS (AB SCIEX QTRAP 5500, Framingham, USA).

### 3.7. Statistical Analysis

PK Solver software was used to determine the pharmacokinetic parameters using a non-compartmental method (version 2.0, China). SPSS software was used for statistical analysis (version 27). The minimal level of significance was set at *p* < 0.05.

## 4. Conclusions

The present study thoroughly investigated the effects of polymer blends on in vitro release/degradation and the pharmacokinetics of MOX-MS. Microspheres prepared using different ratios of low- and high-molecular-weight PLGA exhibited similar surface morphologies with wrinkles, drug loading and particle size. The results of the in vitro release and degradation experiments reveal that the release mechanism of microspheres was a combination of pore diffusion and erosion mechanisms. With the degradation of microspheres from the inside to the outside, the autocatalysis of PLGA played a key role in the degradation process. Specifically, the addition of low-molecular-weight PLGA significantly shortened the lag phase. Pharmacokinetics results demonstrate that moxidectin microspheres could be released for 60 days with a small blood drug concentration fluctuation. F3, prepared using high-molecular-weight PLGA (75/25, 70 kDa) and low-molecular-weight PLGA (50/50, 23 kDa) at a ratio of 2:1, was considered to be the optimal formulation due to its highest AUC (AUC_0-t_ = 596.53 ng/mL·d) and the steadiest plasma concentration compared to other formulations. Overall, the knowledge from the studied formulations may provide a basis for using polymer blends to eliminate the lag phase of hydrophobic drug-loaded PLGA microspheres.

## Figures and Tables

**Figure 1 ijms-24-14729-f001:**
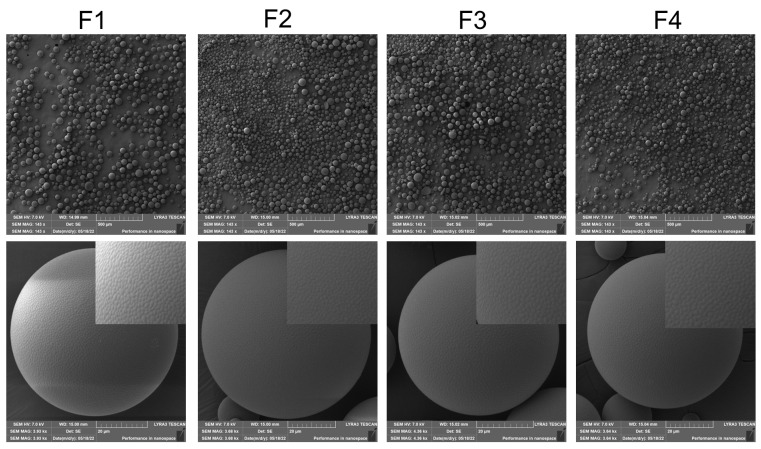
SEM images of MOX-MS.

**Figure 2 ijms-24-14729-f002:**
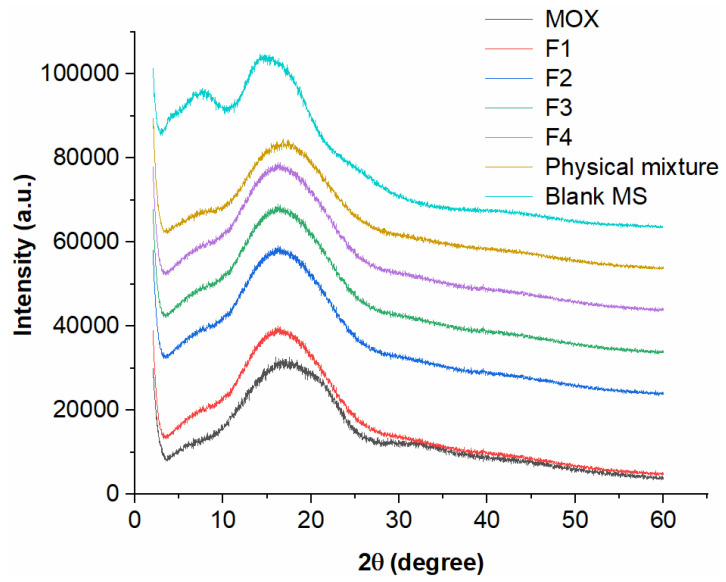
PXRD curves of moxidectin, blank microspheres, MOX-MS (F1–F4), physical mixture of moxidectin and blank microspheres.

**Figure 3 ijms-24-14729-f003:**
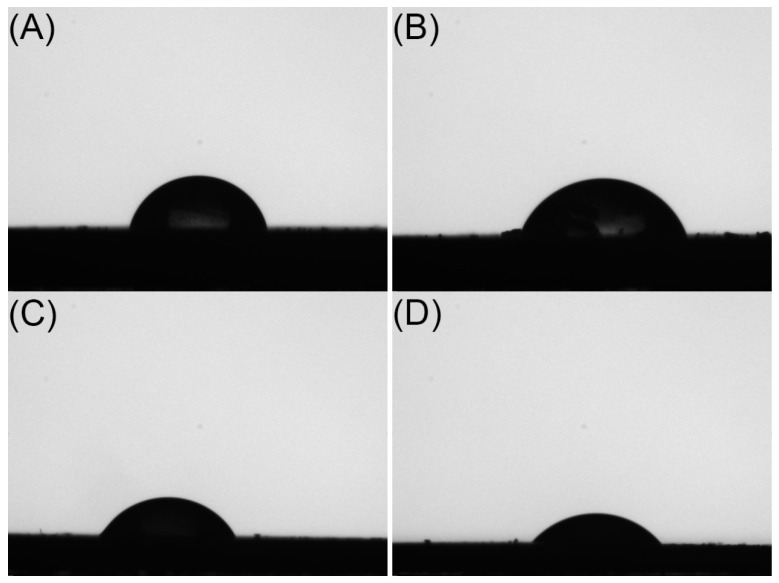
Shadow images of water droplets on F1 (**A**), F2 (**B**), F3 (**C**) and F4 (D).

**Figure 4 ijms-24-14729-f004:**
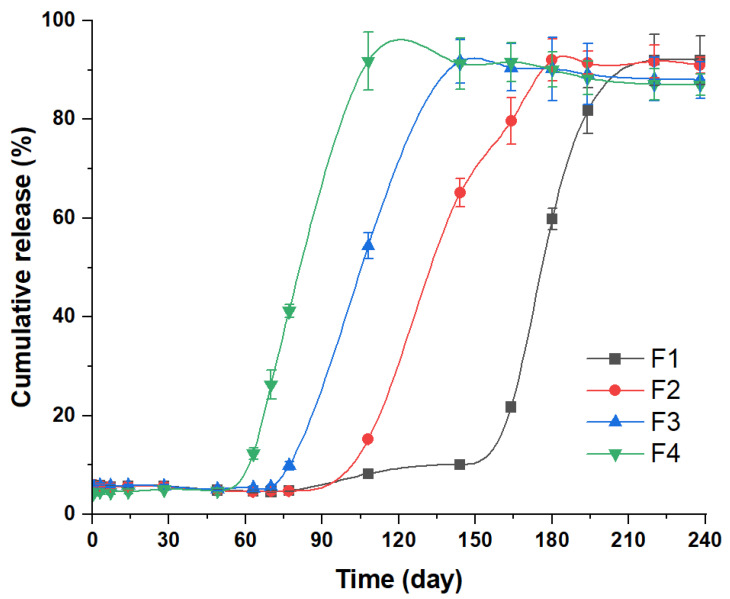
In vitro release profile of MOX-MS in 10 mM PBS solution (pH 7.4, containing 0.5% SDS and 0.02% NaN_3_) with a shaking speed of 100 rpm at 37 °C (the ratio of PLGA (75/25, 75 kDa) to PLGA (50/50, 23 kDa): F1 = 1:0; F2 = 9:1; F3 = 2:1; F4 = 1:1).

**Figure 5 ijms-24-14729-f005:**
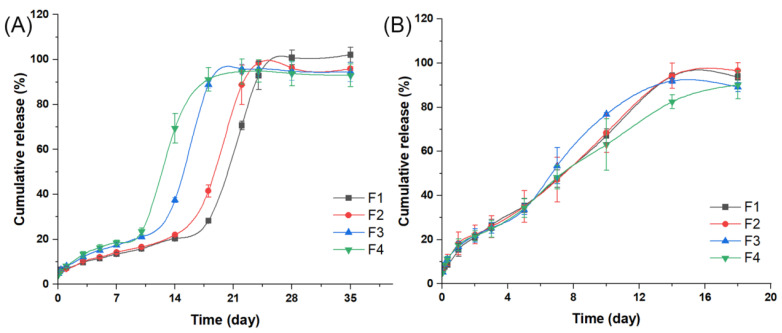
The accelerated in vitro release profiles of MOX-MS at 50 °C (**A**) and at 60 °C (**B**).

**Figure 6 ijms-24-14729-f006:**
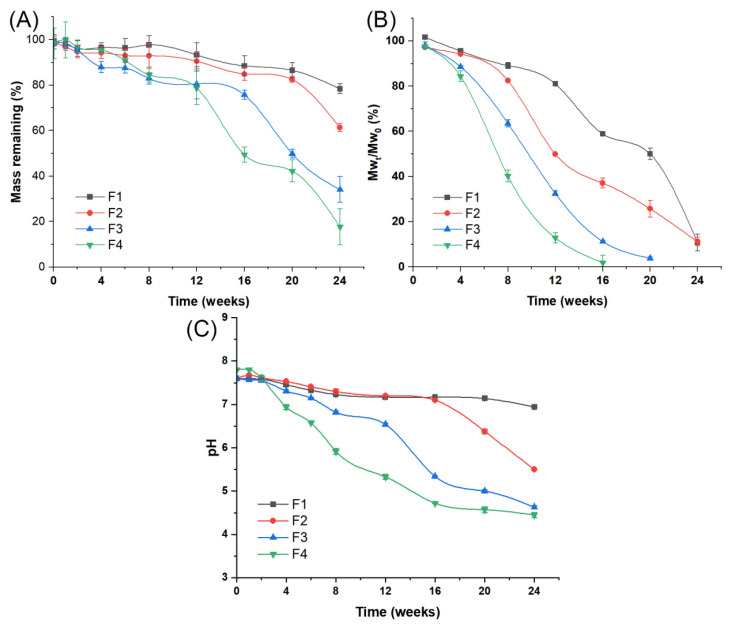
The in vitro degradation process of MOX-MS incubated in 10 mM PBS solution (pH 7.4, containing 0.5% SDS and 0.02%NaN_3_) at 37 °C: (**A**) mass changes of MOX-MS; (**B**) polymer Mw changes of MOX-MS; (**C**) pH changes of release medium.

**Figure 7 ijms-24-14729-f007:**
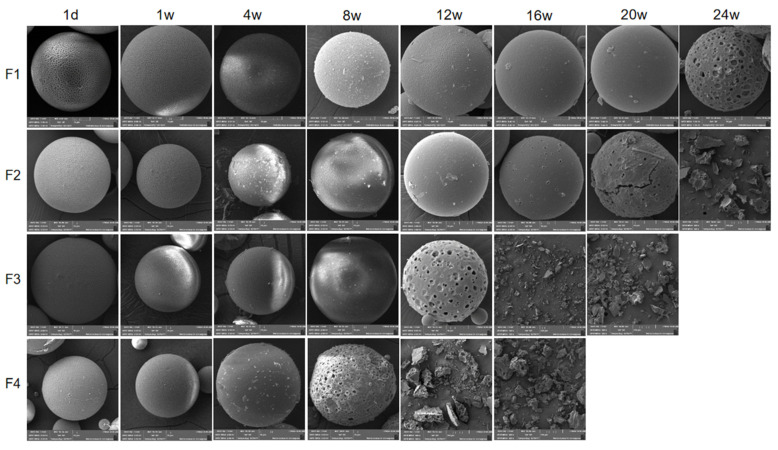
SEM images of MOX-MS during different periods in vitro.

**Figure 8 ijms-24-14729-f008:**
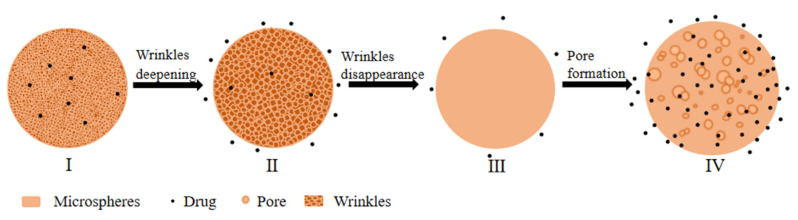
Schematic illustration of the in vitro release mechanism of MOX-MS. From I to II: Drugs on the surface of the microspheres diffused into the release medium formed the initial burst release. From II to III: Wrinkles on the surface of the microspheres gradually disappeared, and only a small amount of drugs was released; thus, the microspheres entered a long lag phase. From III to IV: Many pores formed, accompanied by a large amount of drug released from microspheres at nearly zero-order and the microspheres entered an erosion-controlled release phase.

**Figure 9 ijms-24-14729-f009:**
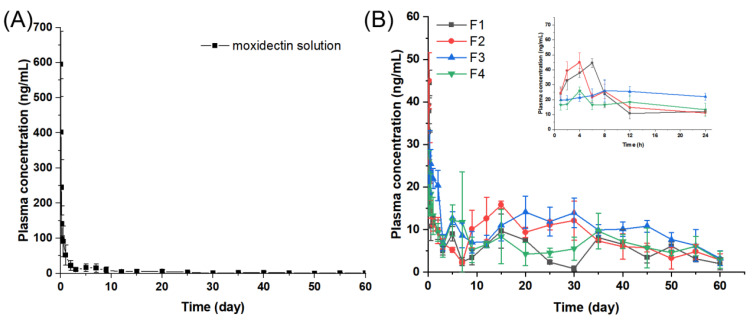
Plasma concentration–time profiles of moxidectin after a single subcutaneous injection of moxidectin solution (**A**) or MOX-MS (**B**) (1 mg/kg). (*n* = 6; mean ± SD).

**Table 1 ijms-24-14729-t001:** Formulation and physicochemical properties of MOX-MS (*n* = 3; mean ± SD).

Formulation	The Proportion of PLGA (75 kDa) to PLGA (23 kDa)	Drug Loading (%)	Encapsulation Efficiency (%)	Mean Particle Size (μm)	Span Value
F1	1:0	35.41 ± 0.26	88.3 ± 0.69	47.723 ± 1.911	1.305 ± 0.097
F2	9:1	36.22 ± 0.14	91.0 ± 0.12	46.899 ± 1.665	1.297 ± 0.071
F3	2:1	36.51 ± 0.21	91.7 ± 0.14	43.371 ± 1.836	1.325 ± 0.077
F4	1:1	36.49 ± 0.21	91.8 ± 0.16	40.784 ± 2.101	1.276 ± 0.081

**Table 2 ijms-24-14729-t002:** Mathematical model fitting results for in vitro release of MOX-MS at 37 °C, 50 °C and 60 °C.

Formulation	37 °C	50 °C	60 °C
	Sigmoidal Equation	Sigmoidal Equation	Zero-Order Equation
F1	Q = 100.04 – 94.56/(1 + e^((t−182.17)/11.64)^) R^2^ = 0.984	Q = 123.64 – 118.63/(1 + e^((t−23.24)/5.00)^) R^2^ = 0.914	Q = 5.52t + 5.97 R^2^ = 0.978
F2	Q = 90.91 – 85.65/(1 + e^((t−131.90)/11.75)^) R^2^ = 0.996	Q = 101.23 – 96.34/(1 + e^((t−17.22)/3.47)^) R^2^ = 0.989	Q = 5.37t + 6.11 R^2^ = 0.977
F3	Q = 89.51 – 84.45/(1 + e^((t−104.99)/8.17)^) R^2^ = 0.980	Q = 110.30 – 104.74/(1 + e^((t−15.63)/3.62)^) R^2^ = 0.930	Q = 6.28t + 6.20 R^2^ = 0.970
F4	Q = 88.60 – 84.05/(1 + e^((t−78.53)/6.54)^) R^2^ = 0.984	Q = 101.61 – 100.13/(1 + e^((t−13.88)/4.29)^) R^2^ = 0.943	Q = 5.57t + 5.23 R^2^ = 0.922

T is the time of drug released from MOX-MS; Q is % of drug released at the time t; R^2^ is the correlation coefficient fitted by mathematical models.

**Table 3 ijms-24-14729-t003:** Mathematical model fitting results for mass changes of MOX-MS, polymer Mw changes of MOX-MS and pH changes of release medium during the in vitro degradation process.

Formulation	Mass Changes of MOX-MS	Polymer Mw Changes of MOX-MS	pH Changes of Release Medium
	Zero-Order Equation	Zero-Order Equation	Zero-Order Equation
F1	W_t_ = −0.78t + 99.26 R^2^ = 0.942	M_t_ = −1.97t + 103.89 R^2^ = 0.948	F_t_ = −0.026t + 7.55 R^2^ = 0.932
F2	W_t_ = −1.16t + 99.51 R^2^ = 0.792	M_t_ = −4.67t + 108.72 R^2^ = 0.927	F_t_ = −0.99t + 8.065 R^2^ = 0.914
F3	W_t_ = −2.20t + 99.71 R^2^ = 0.937	M_t_ = −5.80t + 108.14 R^2^ = 0.964	F_t_ = −0.14t + 7.78 R^2^ = 0.963
F4	W_t_ = −2.56t + 105.67 R^2^ = 0.943	M_t_ = −5.80t + 108.14 R^2^ = 0.980	F_t_ = −0.20t + 7.90 R^2^ = 0.990

t is the degradation time of MOX-MS; W_t_ is the weight of MOX-MS after degradation for t days; M_t_ is the weight-average molecular weight of MOX-MS after degradation for t days; F_t_ is the pH values of the release medium after degradation for t days; R^2^ is the correlation coefficient fitted by mathematical models.

**Table 4 ijms-24-14729-t004:** The non-compartmental model parameters of moxidectin after a single subcutaneous injection of MOX-MS and moxidectin solution into rats at a dose of 1 mg/kg. (*n* = 6; mean ± SD).

Parameter	Moxidectin Solution	F1	F2	F3	F4
C_max_ (ng/mL)	595.65 ± 92.64	44.49 ± 2.98	45.06 ± 6.47	28.30 ± 4.60	27.25 ± 4.33
T_max_ (d)	0.04 ± 0.00	0.24 ± 0.03	0.15 ± 0.03	0.72 ± 0.68	0.17 ± 0.00
T_1/2_ (d)	8.66 ± 2.35	12.52 ± 3.02	18.66 ± 3.15	17.72 ± 5.26	24.55 ± 9.14
AUC_0–1d_ (ng/mL·d)	135.85 ± 28.51	19.54 ± 2.68	19.91 ± 2.23	22.98 ± 2.29	16.80 ± 3.28
AUC_0–t_ (ng/mL·d)	424.64 ± 84.54	331.78 ± 33.78	494.85 ± 67.09	596.53 ± 39.39	407.93 ± 66.77
AUC_0–∞_ (ng/mL·d)	434.81 ± 87.64	369.20 ± 58.81	569.87 ± 54.05	667.06 ± 94.20	437.73 ± 77.36

C_max_, maximum concentration; T_max_, time to reach C_max_; T_1/2_, half-life; AUC_0–1d_, area under the curve from zero to day 1; AUC_0–t_, area under the curve from zero to the last measurable plasma concentration; AUC_0–∞_, area under the curve from zero to infinity.

**Table 5 ijms-24-14729-t005:** Mean plasma concentration of moxidectin after a single subcutaneous injection of moxidectin solution and MOX-MS into rats (1 mg/kg) (mean ± SD; *n* = 6).

Time	Mean Plasma Concentration (ng/mL)				
	Moxidectin Solution	F1	F2	F3	F4
1–60 days	9.06	5.78	8.02	10.81	7.24
1–30 days	13.30	6.26	9.65	12.34	7.90
30–60 days	1.18	4.33	6.04	8.84	5.94

## Data Availability

The original contributions presented in the study are included in the article and Appendix A; further inquiries can be directed to the corresponding author.

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
