# Peer review of "The Effect of Polymer Blends on the In Vitro Release/Degradation and Pharmacokinetics of Moxidectin-Loaded PLGA Microspheres"

_ijms, 2023, doi:10.3390/ijms241914729_

Round 1

Reviewer 1 Report

The paper entitled “The Effect of Polymer Blends on the In Vitro Release/Degradation and Pharmacokinetics of Moxidectin-loaded PLGA Microspheres” can be considered for publication after major revision.

11.     Authors should review all the uncertainty values given in this manuscript as well as the standard deviation values. One cannot accept a value such as 523.57 ± 151.66 (ng/mL·d), this is not admissible. More realistic values with SD have to be given, more particularly in Table 2.

22.     The theoretical kinetic laws of the release profile, the accelerated in vitro, the in vitro degradation process, the mass changes and the pH changes of release medium of MOX-MS should be deduced (See figures 3, 4 and 5). From the results presented on the various Figures, authors should deduce the approximated kinetic law. Please do not give broken curves. Give rather smooth and continuous and derivable curves.

33.     The information on the dispersive surface energy, the polar and specific interactions and Lewis’s acid-base of solid have to be determined. These physicochemical properties are very useful to more understand the behavior of polymer blends on the in vitro release/degradation and pharmacokinetics of moxidectin-loaded PLGA microspheres.

44.     The role of pH on the thermodynamic and kinetic of the in vitro release/degradation should be discussed more deeply in terms of kinetic laws.

55.     Equations 1 and 2 should be rewritten. Please give equations with mathematical symbols and not phrases like Weight of MOX in microspheres⁄Tolal weight of microspheres or Actual MOX loading⁄Theoretical MOX loading. This is not so easy to be readable.

Reviewer 2 Report

The present manuscript presents scientifically significant investigations on the effect of polymer blends on the in vitro release/degradation and pharmacokinetics of moxidectin-loaded PLGA microspheres. The obtained results could be valuable for the pharmaceutical technology and medical sciences. The following remarks were noticed:

1. There are a few English grammar and style inaccuracies.

2. A formula for calculation of the cumulative release has to be added?

3. The authors have to explain why the release medium contained SDS and NaN3 and why the in vitro lesease experimewnts were not conducted in simulated gastrointestinal medium?

4. The study would benefit if the experimental release results are described by appropriate mathematical models.

The present manuscript presents scientifically significant investigations on the effect of polymer blends on the in vitro release/degradation and pharmacokinetics of moxidectin-loaded PLGA microspheres. The obtained results could be valuable for the pharmaceutical technology and medical sciences. The following remarks were noticed:

1. There are a few English grammar and style inaccuracies.

2. A formula for calculation of the cumulative release has to be added?

3. The authors have to explain why the release medium contained SDS and NaN3 and why the in vitro lesease experimewnts were not conducted in simulated gastrointestinal medium?

4. The study would benefit if the experimental release results are described by appropriate mathematical models.

Round 2

Reviewer 1 Report

Accepted